# Live Fluorescence Imaging of F-Actin Organization in Chick Whole Embryo Cultures Using SiR-Actin

**DOI:** 10.3390/cells10071578

**Published:** 2021-06-22

**Authors:** Manuel Schmitz-Elbers, Gražvydas Lukinavičius, Theodoor H. Smit

**Affiliations:** 1Department of Orthopaedic Surgery, Amsterdam University Medical Centres, Amsterdam Movement Sciences, 1105 AZ Amsterdam, The Netherlands; manuel.schmitz-elbers@iwm.fraunhofer.de; 2Chromatin Labeling and Imaging Group, Department of NanoBiophotonics, Max Planck Institute for Biophysical Chemistry, 37077 Göttingen, Germany; grazvydas.lukinavicius@mpibpc.mpg.de; 3Department of Medical Biology, Amsterdam University Medical Centres, 1105 AZ Amsterdam, The Netherlands

**Keywords:** live fluorescence imaging, whole embryo culture, SiR-actin, F-actin

## Abstract

Morphogenesis is a continuous process of pattern formation so complex that it requires in vivo monitoring for better understanding. Changes in tissue shape are initiated at the cellular level, where dynamic intracellular F-actin networks determine the shape and motility of cells, influence differentiation and cytokinesis and mediate mechanical signaling. Here, we stain F-actin with the fluorogenic probe SiR-actin for live fluorescence imaging of whole chick embryos. We found that 50 nM SiR-actin in the culture medium is a safe and effective concentration for this purpose, as it provides high labeling density without inducing morphological malformations.

## 1. Introduction

Embryogenesis is a complex process of tissue formation driven by the tightly coordinated behavior of numerous cells. Next to proliferation and programmed cell death, this involves shape changes and relative cell movements [1]. Cell shape and motility as well as mechanotransduction, differentiation, and cytokinesis are mainly determined by the architecture and contractility of the intracellular filamentous actin (F-actin) network [2,3,4]. In fixated samples, F-actin is often visualized by fluorescently labeled phalloidin, a filament-binding toxin of low molecular weight assumed to “provide the most complete and accurate picture of the actin cytoskeleton” [5,6,7]. Phalloidin does not have the inherent disadvantages of antibody-based labeling techniques, such as non-specific binding, actin scavenging, high background, or epitope variations between actin species [8,9]. However, to understand the role of F-actin in morphogenesis, it is necessary to study it dynamically in vivo. In principle, actin dynamics can be visualized in vivo by microinjecting fluorescently labeled actin and phalloidin; by “bead loading” [10,11,12]; or by transfection with genetically encoded actin reporters like Lifeact and Green Fluorescent Protein actin (GFP-actin) [13]. Each of these approaches, however, comes with disadvantages: microinjection of fluorescently labeled actin is technically demanding, and phalloidin disturbs actin dynamics by filament stabilization. Furthermore, microinjection is low throughput and involves a temporal disruption of the cell membrane. Moreover, bead loading relies on temporal mechanical disruption of the cell membrane and is only applicable to monolayers [12]. Genetically encoded actin reporters turned out to be biased in their staining of different F-actin species [7].

Recently, a powerful probe was introduced to overcome the hurdles in live fluorescent imaging of actin dynamics. Silicon-rhodamine actin (SiR-actin) [14] combines the fluorophore silicon-rhodamine (SiR) [15] with a synthetic and less cytotoxic derivative of jasplakinolide [16], a naturally occurring cyclic peptide known to bind to F-actin [17]. Unlike fluorescent phalloidin derivatives, SiR-actin is cell permeable in vivo and only minimally disturbs the cell’s actin dynamics. With excitation and emission in the near-infrared spectrum, the probe is highly biocompatible, increasing its fluorescence intensity 100-fold upon binding to F-actin in vitro [14]. Finally, SiR-actin provides a high signal-to-noise ratio, without the need of washing before in vivo imaging. So far, SiR actin has been successfully used in high- and super-resolution imaging. For example, it helped to reveal the ubiquitous subcortical cytoskeleton periodicity in the nervous system [18,19] and the dynamic nature of the actin cytoskeleton in mammalian red blood cells [20,21]. The list of possible applications continues to increase, but SiR-actin has not been successfully used for live fluorescent imaging of F-actin in whole embryo cultures.

We could successfully fill this gap by adding SiR-actin to the culture medium of early chick embryos grown in the “submerged filter paper sandwich” [22], a variant of the well-established filter paper carrier technique [23] (Figure 1). As the medium surrounds the embryo during the entire experiment, our idea was that it could function as a reservoir for the fluorogenic SiR-actin throughout the embryo’s development and allow the dye to bind to F-actin without further manipulations or washing.

We used SiR-actin to image the supra-cellular organization of F-actin during somite formation in the chick embryo. Somites are the predecessors of vertebrae, intervertebral discs, ribs and muscles and impose a segmented organization on the peripheral nervous system [24]. The characteristic structural feature of a somite is an apical F-actin ring that connects the epithelial outer cells of the somite [25,26]. The contractility of this structure is presumably responsible for the spherical shape of somites [27]. By trial and error, we found that a SiR-actin concentration of 50 nM is both safe and effective for live fluorescent imaging of early chick embryos: it provides high labeling density without the occurrence of morphological abnormalities, such as head malformations or incomplete somite separations that may be caused by interference of the fluorescent probe with F-actin dynamics.

## 2. Materials and Methods

### 2.1. Culture Medium

We prepared the culture medium as a mixture of Pannett-Compton (PC) -saline [28,29] and freshly harvested thin albumen as previously described [22]. We reduced the thin albumen percentage from 40 to 2% because preliminary experiments had shown that SiR-actin uptake into the embryonic tissue was restrained in presence of high concentrations. Presumably, most SiR-actin molecules retained in the culture medium by non-specific interaction with proteins, make up about 10% of the thin albumen volume in the egg [30] and about 4% of the volume in our original culture medium. Here, Ovalbumin (54%), Ovotransferrin (12%) and Ovomucoid (11%) are the most prevalent proteins. The reduction of thin albumen percentage to 2% did not impair embryonic development. Penicillin/Streptomycin (10,000 U/mL) in 100× dilution were added to prevent bacterial infections. SiR-actin was purchased from tebu-bio BV, Heerhugowaard (NL). It was stored as 1 mM stock solution with anhydrous dimethyl sulfoxide (DMSO) at −20 °C and, after thawing, added to the culture medium in varying concentrations.

### 2.2. Submerged Filter Paper Sandwich (Figure 1)

Fertilized chicken eggs (*Gallus gallus*, Drost BV, Loosdrecht, North Holland, The Netherlands) were incubated at 37.5 °C in a humidified atmosphere for 32–36 h and subsequently transferred into submerged filter paper sandwiches [22]. In short, embryos were clamped between two small sheets of thick filter paper with a central aperture. The filter paper surrounded the embryo like a picture frame from both sides. Explants were cultured in 6 cm culture dishes, stabilized by stainless steel rings and kept fully submerged in the culture medium during the whole experiment [22]. Where possible, the culture medium was covered with a layer of light mineral oil to prevent evaporation.

### 2.3. SiR-Actin Live Fluorescence Microscopy (Figure 2)

Long-term darkfield microscopy and widefield fluorescent time-lapse imaging were achieved by using a long working distance, upright zoom microscope (Zeiss Axio Zoom V.16; Carl Zeiss Microscopy, Jena, Germany), equipped with a PlanNeoFluar Z 1.0× objective, a Cy-5 filter cube (excitation wavelength filter 625–655 nm, emission wavelength filter 665–715 nm) and a Hamamatsu ORCA Flash 4.0 camera (all purchased from Carl Zeiss BV, ZEISS Group, Breda, Netherlands). Time lapse images were acquired at 5 min intervals. Sixteen-bit grayscale images of 2048 × 2048 pixels were acquired at different magnifications of the zoom objective. The tiles module of the Zeiss Zen software was used to acquire panoramic images with a 10% overlap of single tiles (ZEN pro 2012, equipped with modules “Tiles/Positions”, “Time Lapse” and “Experiment Designer”, was purchased from Carl Zeiss BV, ZEISS Group, Breda (NL)). Single tiles were stitched afterwards using the Zen software.

### 2.4. SiR-Actin and Phalloidin Co-Staining in Fixated Samples (Figure 3A)

Embryos were transferred into filter paper sandwiches [22] and carefully washed in phosphate-buffered saline (PBS). Then, embryos (still in filter paper sandwiches) were fixated overnight in 4% paraformaldehyde (PFA) in PBS at 4 °C. After repeated washing steps in PBS (3 × 5 min), embryos were permeabilized in 1% Triton-X-100 in PBS for minimum 1.5 hrs. After washing in PBS (3 × 5 min), embryos were cut out of the filter paper carriers, transferred to 1.5 mL Eppendorf tubes and incubated for 1.5 h with 250 nM SiR-actin and 250 nM phalloidin (Alexa Fluor™ 555 Phalloidin, Thermo Fisher Scientific, Landsmeer (NL)), and 0.1% Triton-X-100 in PBS. After washing (3 × 5 min), embryos were mounted on microscope slides with a drop of mounting solution (ProLong™ Gold Antifade Mountant, Thermo Fisher Scientific, Landsmeer (NL)) and sealed with a cover slip. A layer of parafilm with a rectangular aperture was used as a spacer between microscope slide and coverslip. Samples were imaged on an inverted spinning disc confocal microscope (Nikon Ti2-E Eclipse + Yokogawa CSU W1 confocal scanner unit), equipped with a Nikon CFI Plan Apochromat Lambda 20× air objective (numerical aperture NA = 0.70) and an Andor iXon-Ultra-888 Back illuminated EMCCD camera. Images were acquired with a depth of 16-bit and had a size of 1024 × 1024 pixels (pixel size 0.65 µm × 0.65 µm). Laser power was set to 10% (561 Cobolt Jive 200 mW and 639 Toptica iBeam Smart 500 mW); a gain of 300 and an exposure time of 50 ms was used for both channels.

## 3. Results

### 3.1. Proof of Principle—Increasing Fluorescence Signal While Development Progresses

Preliminary experiments with SiR-actin concentrations ranging from 3 nM to 450 nM in the presence of 2% thin albumen revealed that a satisfying fluorescence signal did not develop for a SiR-actin concentration below 30 nM. On the other hand, SiR-actin concentrations higher than 200 nM led to morphological anomalies during later embryonic development. Figure 2 shows selected time-lapse frames of two chicken embryos, cultured ex ovo as submerged filter paper sandwiches [22] in the presence of 50 and 240 nM SiR-actin. A young embryo, starting from the 1–3-somite-stage (HH7 - 8 [31], Figure 2A,B) and cultured in the presence of 240 nM SiR-actin, progressed to the 10-somite stage (Hamburger-Hamilton stage 10 (HH10)) over 12 h of incubation (Movie S1_Young embryo 240 nM SiR-actin and Appendix A). As expected, the SiR-actin signal continuously increased in strength. The staining started with a spotted appearance of medial parts of the early embryonic heart, probably due to its high morphogenetic activity and its accessibility for the dye. Then, the embryonic midline with the highly contractile cells of the closing neural tube became visible, along with the remains of the primitive streak. Finally, the somites, segmenting from the presomitic mesoderm on both sides of the embryonic midline, were stained specifically for F-actin, indicating that it takes about 5–7 h for sufficient amounts of SiR-actin to penetrate approximately 100 µm deep into the tissue (Figure 2B). Pre-incubation overnight, in the presence of 240 nM SiR-actin, allowed to acquire time-lapse images of somite formation at higher resolution (Movie S2_240 nM SiR-actin high magnification). Here we could observe the gradual formation of the apical actin cortex in the newly forming somites, starting from the posterior and progressing anteriorly.

### 3.2. Identification of a Safe and Effective Sir-Actin Concentration

While the initial live recordings of F-actin in the young chick embryo were successful, we found that culturing in the presence of 240 nM SiR-actin for more than 12 h induced morphological abnormalities. Incompletely separated somites (white asterisk in Figure 2I), indicate that SiR-actin can disturb the actin machinery; this is commensurate with a similar study performed in cultured cells [14]. Morphological abnormalities were neither observed in the control embryos (*n* = 3; thin albumen percentage: 2%), nor in embryos cultured with the submerged filter paper sandwich technique [22]. A slightly older embryo (8-somite stage, HH9 to 9+), incubated with a considerably reduced SiR-actin concentration (50 nM) for more than 28 h did not develop any malformations (Figure 2D,E, Movie S3_older embryo 50 nM SiR-actin, Appendix A). In fact, comparison of average somite formation time between experimental embryos and controls did not indicate any impairment of developmental speed (79 ± 7 min in presence of 50 nM SiR-actin and 86 ± 5 min in controls, *n* = 3 for each, average ± SD, Appendix A).

The SiR-actin signal at 50 nM was hardly visible at lower magnification (Figure 2E). However, image acquisition at higher magnification, and therefore higher numerical aperture of the zoom objective used (change in NA from 0.12 to 0.24, see Appendix A), revealed an intense and highly specific staining of somitic F-actin (Figure 2K).

To compare the development of the SiR-actin fluorescence signal at the two different concentrations, we acquired line intensity profile plots (along white dashed lines in Figure 2B,E) and determined the integrated fluorescence intensity originating from the embryonic neural tube at different time points. Figure 2G presents the results as normalized intensities. The fluorescence signal behaves very similarly for both SiR-actin concentrations: an initial linear increase of fluorescence leads to a peak period, followed by a gradual decrease. Linear fitting of the initial incubation phase shows that the fluorescence increases much quicker with 240 nM SiR-actin (ca. 21%/h) than with 50 nM SiR-actin (ca. 7%/h). During the decrease phase, the fluorescence intensity drops to 60–70% of the maximum value. This could be due to the overall aging and thickening of the embryonic tissue at the selected region, causing additional scattering of the fluorescent signal. We also determined the relation between fluorescent signal and background by calculating the ratio between integrated fluorescent intensities of the vertebral tube (Figure 2C,F) and averaged background value (dashed cyan line in Figure 2B,E) for both SiR-actin concentrations. Figure 2H shows that the signal-to-background ratio (SBR) increases steadily for both concentrations over the first hours of incubation. However, while the SBR reaches a maximum for 240 nM SiR-actin after approx. 6 h of incubation and decreases thereafter, it keeps increasing for 28 h of incubation with 50 nM SiR-actin. Higher magnification imaging results in similar signal and background intensities for the two SiR-actin concentrations (Figure 2J,L). Therefore, preference should be given to lower concentrations to avoid malformations (Figure 2I).

### 3.3. Confocal Imaging of Whole Embryos in Fixated Samples and In Vivo

To compare the staining characteristics of SiR-actin and fluorescently labeled phalloidin, we performed a co-staining of fixated chicken embryos with the same concentration of both probes (250 nM). We observed an almost perfect overlap between the two staining patterns (Figure 3A), which was also confirmed by a colocalization analysis using the Coloc 2 plugin for ImageJ (https://imagej.net/plugins/coloc-2, accessed on 9 June 2021). Here we received a Pearson’s coefficient R = 0.93, indicating a very high degree of correlation between the two signals. This confirms the usefulness of SiR-actin to visualize actin dynamics in vivo. To explore the potential of our culture technique, we incubated live embryos with 50 nM SiR-actin overnight and imaged them, subsequently, on an upright confocal microscope. Figure 3B,C show representative micrographs, illustrating the performance of SiR-actin in labeling F-actin in live chick whole embryo cultures. We could image the F-actin localization during somite formation in overview images and at cellular resolution (Figure 3B,D), as well as the F-actin cortices of endodermal cells (Figure 3C). In particular, the difference in F-actin polarization between the anterior and the posterior part of somite S1, the somite that has just separated from the anterior tip of the unsegmented mesoderm, could be visualized (Figure 3D).

## 4. Discussion

We established a simple protocol for the live staining of F-actin in whole embryo cultures of early chick embryos using SiR-actin in combination with the ‘submerged filter paper sandwich’ [22]. In this approach, the culture medium, surrounding the embryos at all times, functions as a reservoir for SiR-actin dye. Direct manipulations of embryos, like microinjection or electroporation, are dispensable for this staining protocol. By reducing the percentage of thin albumen in the culture medium, we successfully minimized its blocking effect on the SiR-actin uptake into the embryonic tissue. Furthermore, we were able to use only small volumes of SiR-actin stock solution, making the protocol very cost effective. Using confocal microscopy, we could confirm that a high signal intensity and a specific staining pattern, identical to the results obtained for fluorescently labeled phalloidin, was achieved. We showed that the SiR-actin concentration could be lowered sufficiently to avoid morphological abnormalities, like incompletely separated somites, while still delivering a good signal-to-noise ratio.

While it is satisfying to see that the adapted SiR-actin staining protocol allows for live imaging of whole chick embryos, the study has some limitations. First, it only presents a proof-of-principle and an indication that an optimal range of concentrations may exist between thresholds of detectability (30 nM) and safety (200 nM). Replication studies with higher sample numbers are needed for further optimization of the protocol. In the direct comparison between 50 and 250 nM SiR-actin concentrations, embryos of different ages were used (8 somite stage vs. 3 somite stage, respectively). The younger age used in the higher concentration, with possibly higher permeability for SiR-actin, could be responsible for the malformations observed at this concentration. On the other hand, the lower concentration (50 nM) appeared to be sufficient for the older and presumably less permeable embryos. That we never observed morphological malformations in the control embryos or our earlier studies [22] shows that an overdose of SiR-actin is responsible for that.

Due to technical limitations, we could not acquire time lapse images at confocal resolution. We are confident that water immersion objectives (ideally 10× or 20×), in combination with light-sheet microscopy, will allow the acquisition of high-resolution time-lapse series. Immersion of the objective into the culture medium, followed by the application of the mineral oil layer, will create a stable setup without evaporation, which should allow the observation of embryonic development over many hours.

We exemplarily showed that SiR-actin penetrates deep enough into the embryonic tissue to stain even the apical actin cortices of the somites. Other embryonic structures, like the early embryonic heart or the epithelial cells of the endoderm (Figure 3C), are even more accessible to the dye and their imaging would suffer less from scattering of the fluorescent signal. Therefore, we think that our protocol holds large potential for the high-resolution live imaging of many morphogenetic events during embryogenesis.

SiR-actin uptake into the tissue may additionally be improved by using the broad-spectrum efflux pump inhibitor Verapamil, supplied with the SiR-actin kit. We avoided the use of Verapamil, as it is known for its teratologic effects on morphogenesis in early chicken embryos [32]; yet further investigations with varying Verapamil concentrations and extensive controls might lead to an even better SiR-actin fluorescence signal. 

## Figures and Tables

**Figure 1 cells-10-01578-f001:**
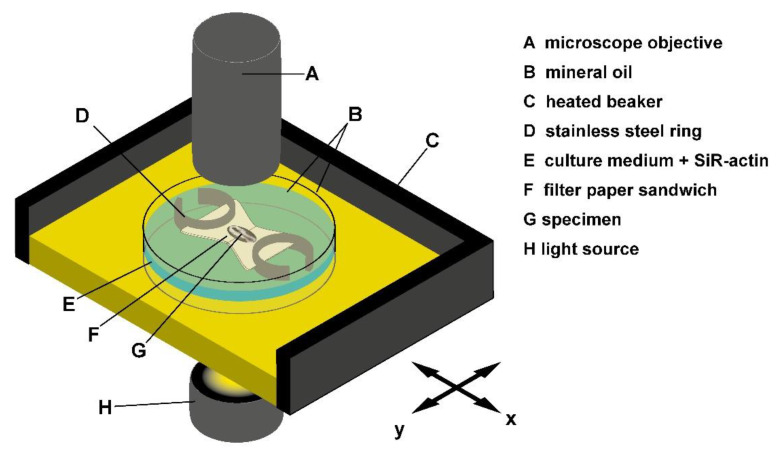
Live fluorescence imaging using the Submerged filter paper sandwich. Schematic view of the experimental setup. The embryo is stabilized by two layers of filter paper and kept fully submerged in the culture medium at all times. SiR-actin can diffuse into the tissue from the culture medium. A layer of light mineral oil prevents evaporation. The heated beaker is filled with mineral oil as well. The sample is illuminated through the glass bottom of the beaker and imaged from above.

**Figure 2 cells-10-01578-f002:**
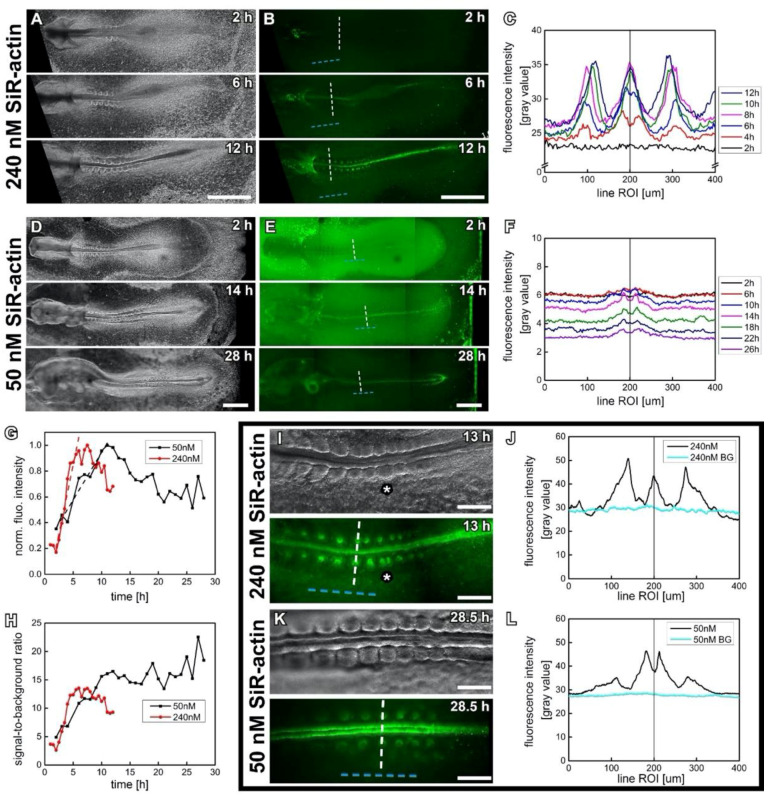
Embryonic development and gradual increase of SiR-actin fluorescence in chick embryos cultured ex ovo, representative images. Time-points indicate incubation time with SiR-actin. Anterior is always to the left. Scale bars, 1 mm (**A**,**B**,**D**,**E**), 250 µm (**I**,**K**). (**A**) Darkfield (DF) microscopic time-lapse frames of 3-somite stage embryo (HH7), cultured in presence of 240 nM SiR-actin. Development progressed up to the 10-somite stage (HH10) during 12 h of incubation. (**B**) Corresponding widefield fluorescence (WF) time-lapse frames of embryo shown in (**A**). Dashed lines indicate line region of interest (ROI) used to measure fluorescence intensities of embryonic midline (white) and surrounding structures (background, cyan). (**C**) Line intensity profile plots across embryonic midline as indicated in (**B**), selected time points. Central peak originates from neural tube fluorescence, lateral peaks from somites. Plots are aligned for the central peak. (**D**) DF images of 8-somite stage embryo (HH9 to 9+), incubated with 50 nM SiR-actin. Development progressed up to the 21-somite stage (HH13-14) over 28 h of incubation. (**E**) Corresponding WF frames of the embryos shown in (**D**). Fluorescence signal builds up slower and less strong than (**B**). (**F**) Line intensity profile plots across the embryonic midline as indicated in (**E**). Plots are aligned to the single peak originating from the fluorescence of neural tube. (**G**) Temporal development of the normalized integrated intensity of fluorescence originating from neural tube at 240 nM and 50 nM SiR-actin concentration. Linear fitting of initial phase shows: fluorescence increases by 21%/h (240 nM) and 7%/h (50 nM). (**H**) Temporal development of signal-to-background ratio (SBR) for different SiR-actin concentrations. SBR was calculated as a ratio between integrated intensity of midline fluorescence and averaged intensity of background fluorescence (cyan dotted line parallel to embryonic midline in (**B**,**E**)). (**I**) Higher magnification DF- and WF-images of the posterior somitic mesoderm of embryo in (**A**,**B**). Incomplete separation of somites indicated (*). (**J**) Line intensity profile plot along dashed lines in (**I**). (**K**) Higher magnification DF- and WF-images of the posterior somitic mesoderm of embryo in (**D**,**E**). Regularly shaped somites and specific F-actin signal visible, showing that 50 nM SiR-actin is sufficient for imaging at higher magnifications. (**L**) Line intensity profile plot along the dashed lines indicated in (**K**), revealing a significantly increased signal strength (compare with (**F**)). Image capture settings: (**A**,**B**) Zoom 2.5, single tile acquired, image size 2048 × 2048 pixels, pixel size 2.6 µm × 2.6 µm, exposure time Cy5 channel 450 ms, darkfield channel 3.5 ms. (**D**,**E**) Zoom 3.1, three tiles (10% overlap) acquired, resulting image size 5735 × 2048 pixels, pixel size 2.08 µm × 2.08 µm, exposure time Cy5 channel 600 ms, darkfield channel 5 ms. (**I**) Zoom 11.2, two tiles (10% overlap) acquired, resulting image size 3880 × 2052 pixels, pixel size 0.580 µm × 0.580 µm, exposure time Cy5 channel 250 ms, darkfield channel 25 ms. (**K**) Zoom 7.8, single tile acquired, image 2048 × 2048 pixels, pixel size 0.832 µm × 0.832 µm, exposure time Cy5 channel 2.61 s, darkfield channel 19.4 ms.

**Figure 3 cells-10-01578-f003:**
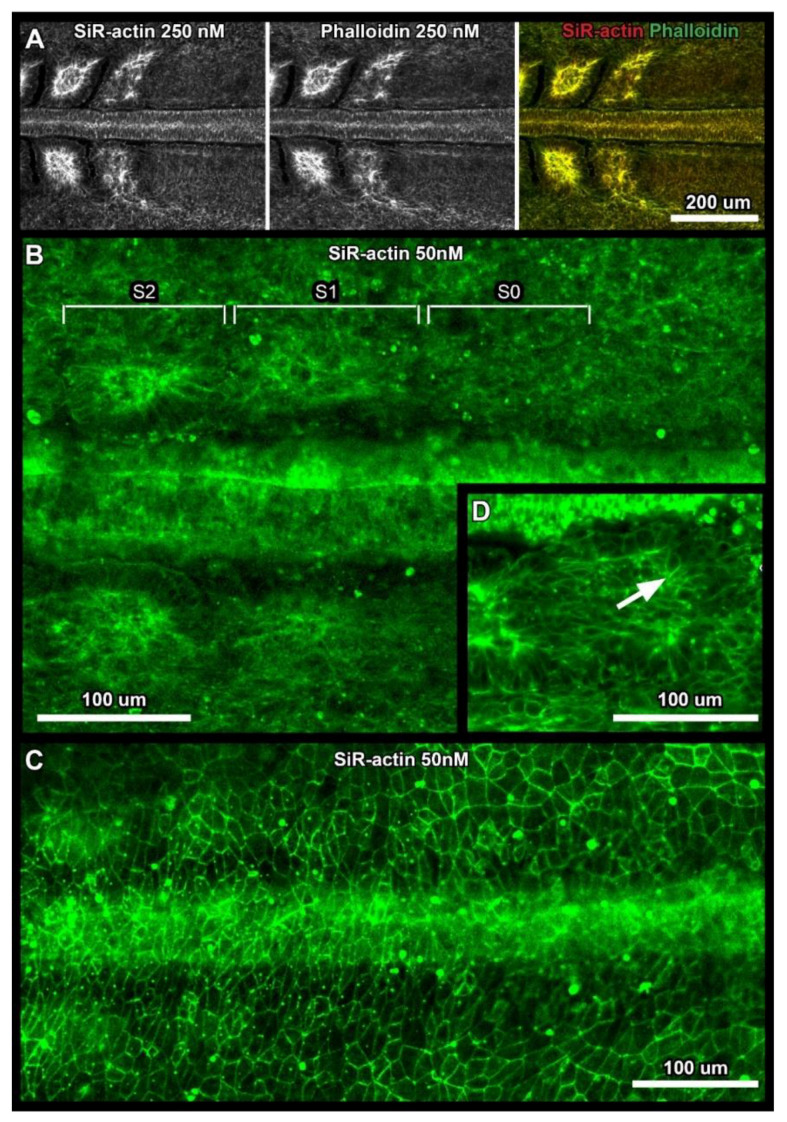
Confocal SiR-actin imaging in fixated samples and in vivo. Representative micrographs, acquired using an inverted spinning disc confocal microscope (**A**) or an upright confocal microscope (**B**–**D**). Anterior is always to the left, ventral view. (**A**) Co-staining of fixated chicken embryos with 250 nM SiR-actin and Alexa-555 conjugated phalloidin showing an almost perfect overlap in staining. (**B**) Overview of segmenting paraxial mesoderm on both sides of the neural tube, in vivo staining with 50 nM SiR-actin. Somite numbers of newly forming somites are indicated according to [27]. (**C**) F-Actin cortices of endodermal cells, in vivo staining. (**D**) High resolution micrograph of live embryo. Formation of the apical actin cortex in somite S1 is more progressed in the posterior half of the somite (white arrow). Image capture settings: (**B**–**D**) Images were acquired using an upright confocal microscope (Leica SP5, Leica Microsystems B.V., Amsterdam (NL)), either using a 10× air objectives (NA = 0.40) or a 63× water immersion objective (NA = 0.90). Due to the short working distance of the objective, images were acquired without a layer mineral oil on top of the medium. Exact settings of the confocal microscope can be found in the Appendix A.

## Data Availability

Not applicable.

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
