# Peer review of "Live Fluorescence Imaging of F-Actin Organization in Chick Whole Embryo Cultures Using SiR-Actin"

_cells, 2021, doi:10.3390/cells10071578_

Round 1

Reviewer 1 Report

The manuscript by Schmitz-Elbers et al. presents a protocol for using a new developed actin probe (SiR-actin) to label F-actin in in chick embryos cultured ex ovo, as well as in fixated and live whole embryos cultures. The authors present data for two different SiR-actin concentrations. Importantly, they showed increased fluorescence with developmental progress. Furthermore, they showed deep penetration of SiR-actin with no morphological abnormalities using 50nM SiR-actin. Using confocal imaging the authors showed very high overlapping between the SiR-actin and the conventional Phalloidin staining, thus further strengthened the specificity of this probe.

I find this manuscript clear and properly controlled. The advances made by the authors over the original publication (Lukinavicius et al. 2014) are quite subtle. Yet, their protocol might be of interest to the scientific community.   

Author Response

We thank Reviewer 1 for his very positive feedback.

No revisions were suggested.

Reviewer 2 Report

In this submission, the authors have used SiR-Actin, a fluorogenic and permeable probe that specifically stains endogenous F-actin, to image whole chicken embryo live cultures. Previously, actin dynamics have been studied in whole vertebrate model using chromobodies, and SiR-actin has been used to stain actin in living cells including neuronal cells.  Here the authors have combined the concepts of aforesaid studies to image actin organization live in whole chicken embryos using SiR-actin.  Moreover, they show that higher levels of SiR-actin can lead to toxicity and malformation of the chicken embryos. However, imaging with lower concentration of the probe could help visualize partial somatogenesis in the chicken embryo. Overall, while the concept is interesting, several critical concerns need to be settled to make the paper appropriate for publication in MDPI cells.

Major points -

1. Although the authors concluded that 50 nM of SiR-actin is an optimal concentration for live imaging, it is not clear how they arrived to this conclusion with only two trial concentrations. Did they perform a gradient check? checked for anomalies at closer changes in probe concentration? Used any antibody markers determining malformations?

2. In Figure 2 it is unclear if the malformations are due to the probe concentration or due to sample prep (tricky prep and also dramatically lower levels of albumin in the prep) because data is shown for just one prep. No n-values included. Did the biological replicates show same effect? Any account for variability across sample preps?

3. In Figure 2 and Figure 3 using verapamil supplied with the SiR-actin kit is a way to enhance the signal by inhibiting efflux pumps and is a practiced method across the field. Did the authors try verapamil to see if that enhances the signal since the signal intensity at the non-toxic concentration is very weak?

4. In Figure 2 it is unclear why two different somite stage embryos were chosen as a starting point for the two different probe concentration? 3-somite for the higher concentration and 8-somite for the lower concentration for the live imaging. Unclear whether the toxicity from probe at higher concentration could also be an effect of the younger tissue? 

5. In Figure 2 Panels J and F looks repetitive to Panels C and F. An inset would be sufficient showing the incomplete separation of the somite. A table showing number of movies analyzed, number/percent of malformed somite would be more informative. Additionally, more still images at more frequent time lapses from the movie at the lower probe concentration would be more insightful.

6. In Figure 3 why do a co-stain with phalloidin and SiR-actin probe with embryos treated with a toxic probe concentration and not otherwise? Intensity plots showing the validity of co-localization of probe and phalloidin not shown? Data for biological replicates unavailable to conclude observation?

Minor points – 

7. Line 16 – remove ‘new’

8. Line 27 – remove ‘but’

9. Line 51-52 – unclear

10. Line 52 – as a

11. Line 95-96 – retained in the culture medium by non-specific

12. Line 100- Remove ‘as shown below’

13. Line 108 – remove ‘thereby’

14. Line 109 – remove ‘petri’; add culture dish

15. Line 114 – substitute for ‘was realized’

16. Line 120 – panoramic

17. Line 122-141 – Include briefly under respective Figure legends

18. Couldn’t find the supplementary data online?

19. Figure 1 – Label the ‘specimen’

Author Response

We thank Reviewer 1 for his very constructive criticism. He makes several valid points which we tried to address as comprehensively as possible.

Major remarks

  1. Although the authors concluded that 50 nM of SiR-actin is an optimal concentration for live imaging, it is not clear how they arrived to this conclusion with only two trial concentrations. Did they perform a gradient check? Checked for anomalies at closer changes in probe concentration? Used any antibody markers determining malformations?

This is an excellent remark, thank you for pointing this out. We did in fact perform preliminary experiments in live chick embryos with concentrations varying from 3 to 450 nM, and found that the signal was insufficient below 30 nM and concentrations higher than 200 nM led to malformations after a certain period of culturing. After the preliminary study we only focused on 50 vs. 250 nM, which we report in this manuscript. Thus, we agree that we cannot conclude that 50 nM is the optimal concentration. Instead, we now claim that 50nM of SiR-actin is a safe and effective concentration for live F-actin staining in whole chick embryos (abstract and line 98, end of introduction). This study is a proof-of-principle to show that SiR-actin can be used to visualize actin organization in the living chicken embryo and our only safety read out was if somitogenesis proceeds similar as in control embryos.

We added the following paragraph in the text (Line 153-158):

“3.1. Proof of principle - increasing fluorescence signal while development progresses

Preliminary experiments with different SiR-actin concentrations ranging from 3 nM to 450 nM in presence of 2 % thin albumen percentage revealed that a satisfying fluorescence signal did not develop for SiR-actin concentration below 30 nM. On the other hand, SiR-actin concentration higher than about 200 nM led to morphological anomalies during embryonic development.”

  1. In Figure 2 it is unclear if the malformations are due to the probe concentration or due to sample prep (tricky prep and also dramatically lower levels of albumin in the prep) because data is shown for just one prep. No n-values included. Did the biological replicates show same effect? Any account for variability across sample preps?

Somite malformations have not been observed for any of the control embryos (n=3) cultured with 2% albumen, nor in the embryos that were cultured with 40 % thin albumen in the culture medium before (see Ref [22] of the manuscript for the original description of the submerged filter paper sandwich). This indicates that the malformation observed was not caused by sample prep or thin albumen concentration, but rather by the high concentration (240 nM) of SiR-actin used. As this was a pilot study, no biological replicates are available, but chapter 3.2 of the results section was adapted to clarify that only one observation per SiR-actin concentration is presented and that none of the controls (n=3) showed similar malformations (line 211-221):

3.2. Identification of a safe and effective SiR-actin concentration

While the initial live recordings of F-actin in the young chick embryo were successful, we found that culturing in the presence of 240 nM SiR-actin for more than 12 hours induced morphological abnormalities. Incompletely separated somites (white asterisk in Figure 2 I), indicate that SiR-actin can disturb the actin machinery; this is commensurate with a similar study performed in cultured cells [14]. Morphological abnormalities were neither observed in the control embryos (n = 3; thin albumen percentage: 2 %), nor in embryos cultured with the submerged filter paper sandwich technique earlier [22]. A slightly older embryo (8-somite stage, HH9 to 9+), incubated with a considerably reduced SiR-actin concentration (50 nM) for more than 28 hours did not develop any malformations (Figure 2 D and E, Movie S3_older embryo 50nM SiR-actin, Figure S2).

  1. In Figure 2 and Figure 3 using verapamil supplied with the SiR-actin kit is a way to enhance the signal by inhibiting efflux pumps and is a practiced method across the field. Did the authors try verapamil to see if that enhances the signal since the signal intensity at the non-toxic concentration is very weak?

This is definitely worth checking in later studies, but we avoided Verapamil here because we did not want to risk additional side effects/malformations. We addressed this comment in the discussion (line 305-309):

“SiR-actin uptake into the tissue may additionally be improved by using the broad-spectrum efflux pump inhibitor Verapamil, supplied with the SiR-actin kit. We avoided the use of Verapamil, as it is known for its teratologic effects on morphogenesis in early chicken embryos [32]; yet further investigations with varying Verapamil concentrations and extensive controls might lead to an even better SiR-actin fluorescence signal.”

  1. In Figure 2 it is unclear why two different somite stage embryos were chosen as a starting point for the two different probe concentration? 3-somite for the higher concentration and 8-somite for the lower concentration for the live imaging. Unclear whether the toxicity from probe at higher concentration could also be an effect of the younger tissue? 

This is a very valid comment that should be addressed by additional experiments in future studies, but it is outside the scope of this study (no additional experiments are possible). SiR-actin can penetrate deeper into the thinner tissues of a young embryo and stains more quickly and strongly. However, the tissue at the anterior tip of the presomitic mesoderm (the not yet segmented part) always has the same (young) developmental age. Mesoderm cells periodically epithelialize in a highly orchestrated way and thus are very sensitive to disturbance of the actin machinery, even if the surrounding tissue becomes stronger. 50 nM SiR-actin appears to be a safe concentration to work with, but an increase to maybe 100 nM might be reasonable to further improve the signal.

We added a comment in the discussion section of the manuscript (line 324-335).

  1. In Figure 2 Panels J and F looks repetitive to Panels C and F. An inset would be sufficient showing the incomplete separation of the somite. A table showing number of movies analyzed, number/percent of malformed somite would be more informative. Additionally, more still images at more frequent time lapses from the movie at the lower probe concentration would be more insightful.

In our eyes, Panels J and L (we think Reviewer 2 means panel L), are not repetitive to Panels C and F because they illustrate that the signal strength compared to background is also very strong for 50 nM SiR-actin if images are acquired at higher magnification (and higher NA). We prepared and additional supplementary figure with more frames from the two time lapse movies.

  1. In Figure 3 why do a co-stain with phalloidin and SiR-actin probe with embryos treated with a toxic probe concentration and not otherwise? Intensity plots showing the validity of co-localization of probe and phalloidin not shown? Data for biological replicates unavailable to conclude observation?

For Figure 3 Panel A we chose a relatively high probe concentration to generate a strong signal in the tissue. The staining presented was prepared on fixated embryos, so there was no risk to disturb actin dynamics. Panels B-D show the results of in-vivo imaging with 50 nM SiR-actin with a very good signal quality.

Instead of preparing intensity plots we performed a colocalization analysis between SiR-actin and phalloidin for validation of the methods. We used the Coloc 2 plugin for FIJI (https://imagej.net/plugins/coloc-2) and received a Pearson´s coefficient R = 0.93, indicating a very high degree of correlation between the two signals. The manuscript was modified accordingly (line 253-257):

“We observed an almost perfect overlap between the two staining patterns (Figure 3 A), which was also confirmed by a colocalization analysis using the Coloc 2 plugin for ImageJ (https://imagej.net/plugins/coloc-2). Here we received a Pearson´s coefficient R = 0.93, indicating a very high degree of correlation between the two signals. This confirms the usefulness of SiR-actin to visualize actin dynamics in vivo.”

Minor remarks

  1. Line 16 – remove ‘new’ – done (line 17)
  2. Line 27 – remove ‘but’ – done (line 26)
  3. Line 51-52 – unclear

Sentence modified for clarification: “Unlike fluorescent phalloidin derivatives, SiR-actin is cell permeable in-vivo and disturbs the cell’s actin dynamics only minimally.“

  1. Line 52 –no longer needed due to textual adjustments.
  2. Line 95-96 – retained in the culture medium by non-specific – corrected (line 87)
  3. Line 100- Remove ‘as shown below’ - done (line 92)
  4. Line 108 – remove ‘thereby’ –done (line 100)
  5. Line 109 – remove ‘petri’; add culture dish –done (line 101)
  6. Line 114 – substitute for ‘was realized’ - done (line 106): exchanged for “was achieved”
  7. Line 120 – panoramic -done (line 112)
  8. Line 122-141 – Include briefly under respective Figure legends –done , and deleted in the manuscript
  9. Couldn’t find the supplementary data online? à will be provided with the revised version.
  10. Figure 1 – Label the ‘specimen’ -done

Reviewer 3 Report

This study conducted by Schmitz-Elbers, et al used fluorogenic probe SiR-actin containing medium to label F-actin in whole chicken embryos for live fluorescence imaging. This study mapped out the optimal concentration of SiR-actin in the culture medium and validated the staining by co-staining with Phalloidin. Here are some concerns that need to be addressed

  1. Line 255 described the Figure 3D, not Fig 2 D
  2. In Result 3.3, why comparing SiR-actin and phalloidin with 250nM rather than 50nM since 50nM is the optimal concentration?
  3. sentences in the introduction should be gone through a grammar check.

Author Response

We thank Reviewer 3 for his very positive feedback.

Remarks

  1. Line 255 described the Figure 3D, not Fig 2 D –agree and corrected

  1. In Result 3.3, why comparing SiR-actin and phalloidin with 250nM rather than 50nM since 50nM is the optimal concentration?

For Figure 3 Panel A we chose a relatively high probe concentration to generate a very strong signal in the tissue. Please remember, that the staining presented here was prepared on fixated embryos, so no risk to disturb actin dynamics.

On the other hand Panels B-D show the results of in-vivo imaging with 50 nM SiR-actin with a very good signal quality.

  1. Sentences in the introduction should be gone through a grammar check a thorough grammar check was done for the whole manuscript. Thanks for pointing this out.